# Process of Work Disability: From Determinants of Sickness Absence Trajectories to Disability Retirement in A Long-Term Follow-Up of Municipal Employees

**DOI:** 10.3390/ijerph18052614

**Published:** 2021-03-05

**Authors:** Päivi Leino-Arjas, Jorma Seitsamo, Clas-Håkan Nygård, Prakash K.C., Subas Neupane

**Affiliations:** 1Finnish Institute of Occupational Health, FI-00250 Helsinki, Finland; Paivi.Leino-Arjas@ttl.fi (P.L.-A.); jseitsamo@gmail.com (J.S.); 2Unit of Health Sciences, Faculty of Social Sciences, Tampere University, FI-33014 Tampere, Finland; clas-hakan.nygard@tuni.fi (C.-H.N.); prakash.kc@tuni.fi (P.K.C.); 3Gerontology Research Center, Tampere University, FI-33014 Tampere, Finland; 4Tampere University Hospital, 33521 Tampere, Finland

**Keywords:** health, lifestyle, registers, survival analysis, trajectory analysis, working conditions

## Abstract

Work disability may originate early during work history and involve sickness absences (SA) and eventually permanent disability. We studied this process over 15 years. Questionnaire data collected in 1981 on health, working conditions, and lifestyle of Finnish municipal employees aged 44–58 years (n = 6257) were linked with registers on SA (≥10 workdays), disability pension, and death from the period 1986–1995. Trajectory analysis was used to assess development in SA (days/year) over 5 years (1981–1985). We analyzed determinants of the trajectories with multinomial regression, while trajectory membership was used as a predictor of disability pension (DP) during the subsequent 10 years in survival analysis. Three SA trajectories emerged: increasing (women: 6.8%; men: 10.2%), moderate (21.2%; 22.7%), and low. In a mutually adjusted model, the increasing trajectory in women was associated with baseline musculoskeletal (MSD), mental and respiratory disorders, injuries, obesity, sleep problems, and low exercise (effect sizes OR > 2), and in men with MSD, sleep problems, smoking, low exercise, and non-satisfaction with management. The moderate trajectory associated with MSD, ‘other somatic disorders’, sleep problems, and awkward work postures in both genders; in women, also overweight, cardiovascular and respiratory morbidity, and (inversely) knowledge-intensive work, and in men, smoking and mental disorders were thus associated. Ten-year risks of DP contrasting increasing vs. low SA were more than 10-fold in both genders and contrasting moderate vs. low SA 3-fold in women and 2-fold in men. These findings emphasize the need for early identification of workers with short-term problems of work ability and interventions regarding lifestyle, health, and working conditions, to help prevent permanent disability.

## 1. Introduction

Sickness absence and premature exit from the workforce due to ill-health pose a considerable burden to individual workers and their employers, the labor market, and social welfare systems. In addition to enabling rest and recovery when ill, repetitive or prolonged sickness absences may have also adverse consequences for the worker in terms of an increased risk of permanent work disability, unemployment, and social exclusion [1,2,3,4] The societal costs of the associated benefits are estimated to be twice those of unemployment, and raising with the ageing of the workforce [5].

The most important medico-legal causes of disability retirement are musculoskeletal and mental disorders [5]. In addition to chronic health problems [6,7], also working conditions [8,9], health-related lifestyle factors such as obesity and low physical activity [10], smoking and alcohol use [11], as well as sleep problems [12,13] add to the risk of disability retirement. 

The influence of these various factors likely begins early in the process of disability which almost invariably involves a period with sickness absence. Indeed, several factors that associate with disability pension have been found to be determinants of sickness absence as well. Musculoskeletal disorders [14], and particularly with regard to longer absences, mental disorders [15,16] are the most common medical causes of sickness absence. In addition, high physical work demands [17,18], psychosocial factors at work [19], and several aspects of health-related lifestyle [20,21,22,23] increase the risk of sickness absence. Low socioeconomic status is associated with the risk of both sickness absence [24,25] and disability retirement [26,27]. However, the entire sequence of influences beginning with work-related hazards, problems of health, or adverse lifestyle factors, and continuing via accrual of sickness absences finally to retirement from work, has rarely been described within one cohort. 

In studies that focus on determinants of disability retirement, detailed attention has rarely been paid to the antecedent sickness absence experience of the employees. Describing the whole entity of sickness absence is complicated by the varying degree of repetitiveness and different lengths of absence periods. Most previous studies have analyzed sickness absences either as a ‘yes or no’ phenomenon or used predetermined cut-points of the duration of a spell [28]. The annual number of sick-leave days was stronger than the duration of a single spell as a predictor of disability pension in a Swedish study [3]. It would be desirable to delineate the development, or change, in absence behavior and find out factors that associate with the different patterns.

In this study, we approached the process of work disability first by examining trajectories of medically certified sickness absences among Finnish municipal employees over a 5-year register-based follow-up and studying factors that were predictive of membership in these, and second, by examining the relationships of the sickness absence trajectories with subsequent disability retirement during a further 10-year period.

## 2. Materials and Methods

### 2.1. Study Sample

The data of the Finnish Longitudinal Study of Ageing Municipal Employees (FLAME) [29] were used. For the study, the 112 largest occupational groups in the municipal sector in Finland were chosen after negotiations with the Municipal Pension Fund and employee and employer organizations. Subjects with at least five-year seniority in their current occupation were targeted. At baseline in 1981, a questionnaire was mailed to 7344 subjects randomly chosen from these workers in all Finnish municipalities. In total 6257 employees, of whom 55.3% were women, responded to the questionnaire. The response rate was 85.2%. The baseline data were collected between December 1980 and March 1981 when the subjects were aged 44 to 58 years. 

Data gathered by questionnaire were linked with information from the national sickness allowance, pension and mortality registers until 2009. The linkages were made using each person’s unique identification code.

The Ethics Committee of the Finnish Institute of Occupational Health approved the study (325270). Ethical clearance for the register linkages was obtained from the national Data Protection Ombudsman. 

### 2.2. Sickness Absence (SA)

Registry-based information of all-cause SA was received from the Social Insurance Institution of Finland and linked with the questionnaire data. In Finland, daily sickness allowance, paid as compensation for loss of income due to temporary incapacity for work, is available after a waiting period of 10 days (the first day of illness and the following 9 working days). Certification by a physician regularly is a prerequisite for such longer sickness absences. Employers often compensate the loss of salary during the waiting period for their employees. The maximum length of sickness allowance is 300 work days for one disease. If work disability continues after this, the person may be considered for the award of a disability pension.

For the purposes of this study, we calculated the annual number of days on sickness allowance (sickness absence days) from 1981 to 1985. The different developmental paths of SA were constructed using trajectory analysis.

### 2.3. Disability Pension Retirement and Mortality

In Finland, disability pension may be awarded to a person whose work ability has decreased by at least three fifths (two fifths for partial disability pension) due to a disease, injury, or handicap for at least a year. If a person is employed in the public sector when work disability begins, consideration is also given to the ability to work in the current position or job in particular. Data on disability pension as well as other pension awards were obtained from the national registers of the Finnish Center for Pensions, which provides complete information of all retirement events. We studied incident disability pension awards during 1986–1995. Mortality data were extracted from the Finnish National Population Register.

### 2.4. Age and Socioeconomic Status

Age was categorized into four groups: 44‒47, 48‒50, 51‒54, and 55‒58 years.

As indicators of the respondent’s socioeconomic status, we used the level of basic education and the family’s income. Basic education was categorized as ‘elementary school not completed’, ‘elementary school completed’, and ‘comprehensive school or higher completed’. Satisfaction with the family income was assessed by the question “How would you assess the income of your family (response alternatives: 1 = very good, 2 = rather good, 3 = satisfactory, 4 = rather poor, 5 = very poor)?” We used a trichotomy of family income: good (1–2), satisfactory, or poor (4–5).

### 2.5. Lifestyle Factors

BMI [weight (kg)/height (m^2^)] at baseline, based on self-reported weight and height, was classified as normal weight/underweight <25, overweight 25–30, and obesity ≥30 kg/m^2^. Smoking was assessed by the question “Do you now smoke regularly; if yes, how many cigarettes per day?” The responses were dichotomized into non-smokers and smokers (≥1 cigarette/day). Leisure-time physical activity (LTPA) was measured using a question on the average frequency of exercise (‘sports, exercise, or other leisure-time activity that lasted for at least 15–20 min and caused you being out of breath’) during the previous year. Five response alternatives were given (‘brisk exercise at least twice a week’, ‘brisk exercise at least once a week’, ‘some form of exercise once a week’, ‘some form of exercise less than once a week’, ‘not engaging in exercise’). We categorized LTPA as at least twice a week, once a week, or less than once a week.

Sleep problems were assessed using three questions with five-point (0–5) response scales: “How easily do you fall asleep?”, “How continuous is your sleep at night?”, and “Have you recently woken up too early unable to fall back asleep?” Missing responses were substituted with the value 2. The responses were summed up and transformed to create a sleep problem score ranging from 0 to 10. This was categorized in tertiles (no, moderate, frequent).

### 2.6. Morbidity

The subjects responded at baseline to the questionnaire item “Please indicate in the list below the injuries and diseases or disorders that you have at the moment. In addition, indicate whether this has been diagnosed or treated by a physician?” The response alternatives included injury (under this, the low back, upper limbs, lower limbs, and ‘other’ separately listed), musculoskeletal disorders (five disorders and the category ‘any other’ were listed), cardiovascular diseases (four disorders and ‘any other’), respiratory diseases (four disorders and ‘any other’), mental disorders (severe or mild given as options), neurological or sensory disorders (two disorders and ‘any other’), gastro-intestinal diseases (four diseases and ‘any other’), genitourinary diseases (one urinary tract infection and ‘any other’), skin diseases (two diseases and ‘any other’), neoplasms (malignant or non-malignant), hormonal and metabolic diseases (three diseases and ‘any other’), and any diseases of the blood, infectious diseases, and congenital anomalies.

We further constructed dichotomies indicating the presence of one or several disorders in the following categories: musculoskeletal disorders, cardiovascular diseases, respiratory diseases, other somatic diseases, mental disorders, and injuries. We used the responses where the disease or disorder was reported as diagnosed or treated by a physician.

### 2.7. Working Conditions

The baseline questionnaire covered several variables concerning work exposures, assessed on a 4- or 5-point Likert scale referring to the frequency of occurrence, degree of impediment, or satisfaction with (items on management) a work factor. These were grouped into sum scores using factor analysis and correlation analysis [30]. Missing values were substituted with the average of the subject’s work profile group (13 groups) [31]. All sum scores were transformed to a scale from 0 to 10 and categorized as tertiles of gender-specific distributions.

We analyzed the relationships with SA trajectories of the following sum scores: physical and chemical environment (dirtiness; dust, smoke, steam etc., risk of accident; noise; vibration; lighting and glare; heat, cold, and changing temperature; dryness; restless environment and noisy people: Cronbach’s alpha coefficient = 0.85), awkward work postures (bent or twisted postures; otherwise poor posture, repetitive movements: alpha = 0.76), physical demands (carrying objects by hand; frequent walking or movement; standing: alpha = 0.76), job control (possibility to plan one’s own work; possibility to influence one’s own work environment; possibility to receive training and update skills; possibility to apply own skills; possibility to learn new things; possibility to receive recognition and esteem: alpha = 0.85), mental demands (complex decision making; decision making under time pressure; accuracy in information processing: alpha = 0.82), and satisfaction with management (co-operation between employer and employees; supervisor’s attitude; work design and management; distribution of information: alpha = 0.79).

### 2.8. Statistical Analyses

Statistical analyses were made in three steps. First, the development in SA during the first five years of follow-up was described by trajectory analysis. Second, the associations of baseline variables with trajectory membership was determined. Third, the relationships of SA trajectories with disability pension retirement during a further 10-year follow-up were assessed. Analyses were stratified by gender.

The annual number of sickness absence days during the first five years of follow-up was modeled by trajectory analysis (PROC TRAJ in SAS), a semiparametric method that identifies latent groups within the data that tend to have a similar profile of development over time [32]. We used the maximum number of subjects in the analyses. The subjects who died or retired from work during the trajectory time were excluded treating them as missing observations. A model based on the censored normal distribution was used. The selection of the optimal model, the number of trajectories, and their shapes, was based on the Bayesian information criterion (BIC) and the average posterior probabilities of group membership (≥0.7 was considered to indicate good model fit). Each person was assigned to the trajectory to which the posterior probability was the highest. The model with the lowest BIC was selected. 

The means and standard deviations of the baseline variables on health, lifestyle, and working conditions were calculated. Associations of the variables with trajectory membership were then analyzed by multinomial logistic regression. Two models were calculated, adjusted for age, and mutually adjusted for all studied variables, including also age and socioeconomic status (basic education and family income).

As the last step, SA trajectory membership as a predictor of disability pension retirement was described with Nelson–Aalen cumulative hazard plots and Cox regression adjusting for age and socioeconomic status. Follow-up began on 1 January 1986 and ended with old age pension retirement, death, or end of follow-up (31 December 1996), whichever came first.

The analyses were made using SAS version 9.1 (SAS Institute Inc., Cary, NC, USA) and Stata version 16.

## 3. Results

### 3.1. Sickness Absence (SA) Trajectories

In the trajectory analysis of the number of SA days during the first five years of follow-up, a three-group model had the best fit in both genders (Figure 1), with two trajectories (1 = low, 2 = intermediate) of a linear shape and one of a quadratic shape (3 = increasing). In both genders, the low trajectory included the majority of the subjects (2028, i.e. 67.1%, of the men and 2685, i.e. 72.0%, of the women). The intermediate group included 491 (22.7%) of the men and 548 (21.2%) of the women. The increasing trajectory group, among whom the number of SA days rapidly accrued over the follow-up, covered 278 (10.2%) of the men and 227 (6.8%) of the women. The mean posterior probabilities of belonging to trajectory group 1, 2, or 3 were 0.88 (SD 0.12), 0.74 (SD 0.15), and 0.87 (SD = 0.14) among men, and 0.88 (SD 0.12), 0.74 (SD 0.16), and 0.92 (SD 0.12) among women. The expected lines show the model-based expected values of the outcome variable, given the group membership and time. The observed lines show the weighted averages of the original measurements over time, using the posterior probabilities of the corresponding group membership as weights. 

### 3.2. Determinants of SA Trajectories

Next, we analyzed the associations of baseline covariates with membership in the SA trajectories. The distributions of these baseline variables by trajectory group are shown in Table 1. Those who belonged to the increasing trajectory were somewhat older, tended to have a lower educational attainment and family income, were more likely obese and smokers, had lower LTPA, and more often reported sleep problems, chronic diseases and injuries than the other groups, among both women and men.

#### 3.2.1. Lifestyle-Related factors

Health-related lifestyle factors were important predictors of SA trajectories in both genders. Among women (Table 2), overweight and obesity as well as moderate and frequent sleep problems were associated with membership in the intermediate SA trajectory as contrasted with the low, when adjusted for age in model 1. When mutually adjusted for all variables in the table and including age and indicators of socioeconomic status among the covariates (model 2), overweight showed an association with the intermediate trajectory (1.3; 1.0–1.7), obesity with the increasing (2.1; 1.3–4.4), and frequent sleep problems with both the intermediate (1.7; 1.3–2.3) and increasing (2.1; 1.1–4.0) trajectory.

Among men (Table 3), smoking, LTPA once a week or less frequently, and moderate and frequent sleep problems were associated with the intermediate and increasing SA trajectories in model 1. In model 2, smoking (1.6; 1.3–2.1) and frequent sleep problems (1.5; 1.1–2.0) were associated with the intermediate trajectory, while smoking (1.6; 1.0–2.5), low LTPA (2.4; 1.2–4.6), and frequent sleep problems (2.3; 1.4–3.9) associated with the increasing trajectory. 

#### 3.2.2. Health problems

Among women in the age-adjusted model 1, all health problems with the exception of mental disorders associated with the intermediate SA trajectory, and all these, including mental disorders, with the increasing trajectory. In the mutually adjusted model 2, musculoskeletal (2.2; 1.7–2.7) and respiratory (1.7; 1.3–2.4) disorders and the category of ‘other somatic diseases’ (1.5; 1.2–2.0) associated with the intermediate trajectory, and musculoskeletal (3.6; 2.0–6.5), respiratory (2.8; 1.5–5.3), and mental (2.5; 1.1–5.9) disorders with the increasing trajectory.

Among men, all categories of health problems were associated with belonging both to the intermediate and the increasing SA trajectory when adjusted for age. When mutually adjusted for all variables in model 2, musculoskeletal (1.7; 1.3–2.2), mental (1.8; 1.1–2.1), and ‘other somatic’ (1.5; 1.2–2.0) disorders associated with the intermediate but only the musculoskeletal (2.2; 1.5–3.6) with the increasing trajectory.

*Injuries.* We found that in women, injuries were associated with the intermediate and the increasing SA trajectory in model 1 and with the increasing trajectory in model 2 (2.0; 1.0–4.0). In men, an association with the intermediate trajectory was observed when adjusted for age, but no associations were seen in model 2.

*Physical workload.* In both genders in model 1, moderate and high exposure to awkward work postures was associated with membership in the intermediate, and high exposure with the increasing SA trajectory.

Among women in model 2, high exposure to awkward postures was associated with the intermediate trajectory (1.6; 1.1–2.2), while no associations with the increasing trajectory were seen. Among men in model 2, moderate (1.8; 1.3–2.6) and high (1.9; 1.3–2.9) exposure associated with the intermediate trajectory. Again, no relationship with the increasing trajectory was seen.

In women, both moderate and high exposure to physical work demands associated with membership in the intermediate and high exposure with the increasing trajectory. In men, high exposure associated with the intermediate and increasing trajectories in model 1. In model 2, no associations were seen in either gender. 

*Physical and chemical exposures.* In men in model 1, moderate and high exposure to physical and chemical hazards associated with belonging to the intermediate and increasing SA trajectories; in women, statistically significant associations were seen only with the intermediate trajectory. In model 2, no associations were observed in either gender.

*Psychosocial factors at work.* Male employees who were highly satisfied with the management at their workplace were in a decreased risk of belonging to the intermediate and high SA trajectories in model 1. High management satisfaction was also inversely associated with membership in the increasing SA trajectory in model 2 (0.5; 0.3–0.9). In women, moderate management satisfaction was inversely associated with the intermediate trajectory in model 1; no other associations were seen.

Compared with those with low job control, women and men with moderate and high job control were in a decreased risk of belonging to the intermediate SA trajectory in model 1. Similarly, both women and men with the high job control were in a decreased risk of belonging to the increasing trajectory in model 1. In model 2, no associations were observed.

Women with moderate and high mental demands at work had a decreased risk of membership in the intermediate SA trajectory, and women and men with high job control of membership in the increasing trajectory in model 1. In model 2, women with high mental demands had a decreased risk of membership in the intermediate trajectory (0.6; 0.5–0.9), while in men, no associations were seen in the mutually adjusted models.

### 3.3. Sickness Absence Trajectories as Determinants of Disability Pension Retirement

As the last step, we analyzed the relationship of the 5-year SA trajectories with disability pension retirement during the subsequent 10 years. There were 1707 cases of disability pension retirement in total, 845 men (621 in the low, 193 in the intermediate, and 31 in the increasing trajectory) and 862 women (596 in the low, 245 in the intermediated, and 21 in the increasing trajectory) during that period. The Nelson–Aalen curves of disability pension retirement by trajectory group, separately for men and women, and censoring for old age pension and death, are shown in Figure 2. The groups differed statistically highly significantly from each other. Already from the beginning of the follow-up, the curves started diverging and the difference grew during the follow-up among both women and men. The 10-year hazard ratios of disability pension (DP) were assessed adjusting for age and indicators of socioeconomic status. Contrasting the increasing against low SA trajectory, the hazard ratios were 10.5 (95% CI 7.1–15.5) in men and 11.8 (95% CI 7.6–18.3) in women, and contrasting intermediate vs. low SA the ratios were 2.0 (95% CI 1.7–2.3) in men and 3.1 (95% CI 2.6–3.6) in women (Table 4).

## 4. Discussion

In this prospective study of municipal employees, we approached the process of disability for work by first examining which factors evaluated at the start of the follow-up were associated with the developmental trajectories of all-cause sickness absence over the next five years, and then studying how belonging to these was predictive of subsequent permanent disability. We found that variables from all studied domains, i.e. somatic and mental health, physical and psychosocial working conditions, and health-related lifestyle, were associated with subsequent sickness absence trajectories, which in turn closely predicted disability retirement. Although there is a lot of evidence on effects of working conditions, health, and health-related lifestyle on sickness absence on one hand [17,19,21,22], and on absenteeism as a predictor of disability on the other [7,8,9,10], we are not aware of directly comparable previous studies within one cohort. 

Analysis of sickness absence involves challenges due to the often repetitive occurrence of absence periods varying in duration. Describing determinants of long-term sickness absence is of particular interest due to their high individual and societal costs. For this, predetermined cut-points of the length of absence according to e.g., insurance and legislative practices have mostly been used, as no standardized definition of long-term sickness absence exists [33]. Instead, we used trajectory analysis which provides advantages in the analysis of longitudinal data by identifying latent sub-groups with a similar evolution of the outcome over time and easily graspable visualization of change [34]. 

For both men and women during the first five years of follow-up, we observed three distinct latent groups that were well separated from each other as indicated by high posterior assignment probabilities. Similar to trajectory analyses of other cohorts [35,36], the largest group consisted of those with low sickness absence over the whole length of the follow-up. The second, or intermediate, trajectory indicated about twenty absence days annually, while the third, though first at a similar level of absence days as the intermediate group, displayed a sharp increase in absenteeism from around twenty to about 160 to 170 days per year by the end of the period. The increasing trajectory was somewhat larger in men (10%) than women (7%), while the intermediate group comprised about a fifth of both men and women. Some earlier studies have found higher absenteeism in women than men [18,37], but this may be limited to self-certified and other shorter absences [37,38].

Previous studies have often focused on sickness absence related to some particular health condition [39,40]. We included health status in several major diagnostic groups as potential determinants of any absenteeism parallel to workplace factors, health-related lifestyle, and basic socio-demographic factors. Our study sample was already in its mid-career at the start of the follow-up. The average age was slightly above 50 years in the constantly low and in the intermediate sickness absence trajectory, and somewhat higher in the increasing one. In this aging cohort with the outcome of longer absences requiring certification by a physician, morbidity was an important determinant. Musculoskeletal disorders showed strong effects on trajectory membership particularly in women, in line with previous studies [36,41]. The role of musculoskeletal problems in women is further underlined by that injuries, mostly to the musculoskeletal system, associated with the increasing trajectory in the full model. In addition, mental disorders (increasing trajectory in women, intermediate in men,), respiratory diseases (women), and other somatic diseases (intermediate trajectory in both genders) associated with sickness absence in the mutually adjusted models. This is in line with a more recent Danish study, which found that back disorders, mental disorders, and cancer were the health problems predicting long-term sickness absences in a two-year register based follow-up of a general population sample [42]. Long sickness absences due to mental problems showed large differences by occupational class with any disability retirement among Finns in 2007–2014 [43].

Health-related lifestyle also displayed clear relationships with absenteeism. Overweight/obesity in women, smoking in men, and low LTPA and frequent sleep problems in both genders associated with the intermediate and/or increasing trajectories in the full model, in semblance with other studies using trajectory analysis [35,36] and those using other methodological approaches [21,44,45,46]. In another cohort of Finnish municipal employees, smoking and high body mass index were the most prominent lifestyle factors linked with medically certified sickness absence when working conditions and socioeconomic status were controlled for [47].

While working conditions were associated with absenteeism in many comparisons adjusted for age, several associations attenuated with further modeling. The relationship of high and moderate occurrence of awkward work postures in men, and of their high occurrence in women, retained in relation to the intermediate trajectory. Our findings are in line with a study in a random sample of Danish employees, which found that 27% of incident long-term (≥8 weeks) sickness absences in women were attributable to bending or twisting the neck or back, and in men a quarter of such absences to working in a standing or squatting posture and manual materials handling, when psychosocial and lifestyle factors were allowed for [48]. Similarly, a long work history of handling heavy loads, use of high hand grip force, prolonged standing or walking, and squatting and kneeling at work predicted a high sickness absence trajectory in the Finnish general population sample [49]. Among employees of the City of Helsinki physical workload, assessed by an index that covered awkward postures, heavy lifting and other components of loading, was linked with all-cause absenteeism of all lengths during a 3-year follow-up [17].

Of the psychosocial work factors, inverse associations of high satisfaction with management at the workplace in men retained in the full model. The result broadly agrees with those from a 2-year register follow-up in Denmark [50], where low levels of supervisor support and predictability were associated with more absence days. In our study job, control associated with the intermediate and increasing trajectories only when age-adjusted. We also found that high mental demands in women was inversely associated with belonging to the intermediate trajectory. This may reflect the accrual of knowledge-intensive tasks in higher socioeconomic groups. On the other hand, low job autonomy and low task complexity were predictive of sickness absence in forest industry in Finland [51]. In the literature, results on the associations of psychosocial work factors with all-cause sickness absence have been variable [52]. In the Danish study [48], psychosocial factors were not associated with long-term sickness absence when mutually adjusted.

Overall, our results are broadly in line with those from a sample representing all Finnish employees followed up for seven years from year 2000 onwards, where the number of annual all-cause sickness absence periods were modeled in trajectory analysis [36]. In that study, musculoskeletal pains and diseases, mental disorders, frequent sleep problems, obesity, smoking, physical workload, job control, and level of education were predictive of belonging to a constantly high and/or an increasing sickness absence trajectory in a multivariable model. The associations were somewhat weaker than ours, perhaps due to different approaches to sickness absence, or model and sample characteristics. A recent study did not, however, find differences comparing six-year sickness absence trajectories among Finnish municipal and industrial employees [53].

A questionnaire follow-up of those who had responded at baseline was made in 1985, when 1% of the original sample had died, 9% were retired, and 5% had changed occupation [54]. Altogether, 28% (1728/6257) of this elderly cohort experienced disability retirement during the latter part of the follow-up. Sickness absence trajectories were closely associated with subsequent disability retirement. Among subjects belonging to the increasing trajectory, which reached an annual average of about 160 absence days, the risk of disability retirement was increased more than 10-fold compared with the low trajectory. The intermediate group with about 20 annual absence days also carried a substantially increased risk of disability, in men, a two-fold and in women, a three-fold risk. Sickness allowance in Finland is provided for a maximum of 300 days, which is partly reflected in the results. Very strong relationships between high or increasing absenteeism and disability have also been found in other studies using somewhat different analytic designs. A pooled 16-year follow-up of Swedish population studies found that 245 annual days on sick-leave amounted to a 50% probability of disability retirement in both genders [3]. A retrospective study of the 10-year sickness allowance histories of all Finnish residents who had been granted a disability pension in 2011 found a sharply increasing trajectory among 29% and an early high trajectory among 21% of the retirees [55].

Among the strengths of our study are the high response rate at baseline, long prospective design, and the use of official registers as the source of data on both sickness absences and retirement, eliminating recall bias. The data covered more than one hundred occupations in the municipal sector. Sickness absence was analyzed as the development of annual number of days. We had information on several aspects of health, lifestyle, and work and used these in multivariable analyses. Some comparisons between genders could also be made. On the other hand, indicators of work exposures were self-reported in the beginning of the follow-up and working conditions may have changed in the municipal sector since the sample was established. Although the core of many jobs in nursing, kitchen work, transport, administration, or education has retained over time [56,57], with e.g., the general technological advancement, there have been changes enough, and we cannot rule out their effects on the studied relationships. In addition, health-related lifestyle has generally improved in Finland over the recent years. Variation in legislation on short and long-term disability benefits restricts the comparability of our findings to countries where the benefit system is largely similar, e.g., the other Nordic countries. We had no access to data on shorter absences than those lasting at least 10 workdays.

To conclude, we observed several predecessors related to health, lifestyle, and work, of the developmental patterns of all-cause sickness absence during 5 years, which were closely predictive of all-cause disability pension retirement over the next 10 years. These findings emphasize the long timeline eventually leading to permanent disability and the need for early identification of workers with short-term problems of work ability to implement interventions, preferably multifaceted, targeting lifestyle, health problems, and working conditions, to help prevent permanent disability [58,59]. In particular, employees in the lower socioeconomic groups, where many of the risk factors of sickness absence and disability cluster, should be focused on.

## Figures and Tables

**Figure 1 ijerph-18-02614-f001:**
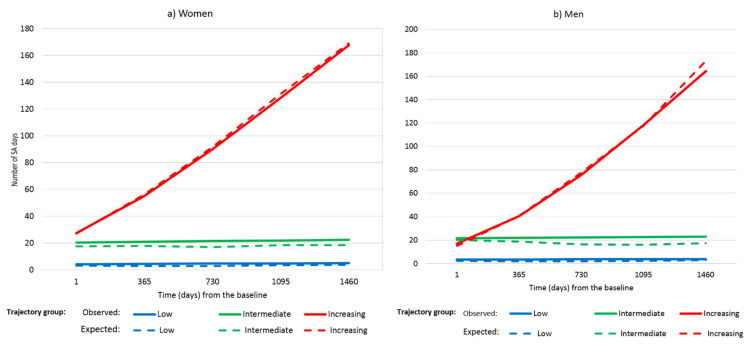
Trajectories of sickness absence (annual number of days) over five years of follow-up (1981–1985) among men and women. The solid line indicates the observed and the dashed line the expected trajectories.

**Figure 2 ijerph-18-02614-f002:**
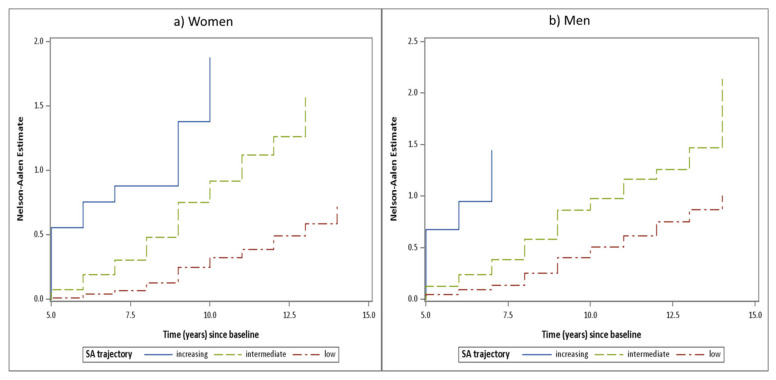
Nelson–Aalen cumulative hazard plots of disability pension retirement during ten years of follow-up (1986–1995) by trajectory of sickness absence during the previous 5 years, censoring for old age retirement and death.

**Table 1 ijerph-18-02614-t001:** Age, socio-economic status, and other characteristics of the sample related to lifestyle, health and work at baseline, by gender and sickness absence trajectory for (**a**) Women and (**b**) Men.

a. Women			
Characteristic	Sickness Absence Trajectory		
	Low (n = 2685)	Intermediate (n = 548)	Increasing (n = 227)
**Age** (Mean, SD)	50.3 (3.6)	50.5 (3.6)	51.7 (3.5)
**Basic education** (%)			
Elementary school not completed	14.5	20.3	25.3
Elementary school completed	47.5	55.8	52.4
Complementary school completed	38.0	23.8	22.2
**Family income** (%)			
Good	40.7	33.8	32.7
Satisfactory	55.5	62.4	62.7
Poor	3.8	3.7	4.6
**Lifestyle factors**			
**Smoking** (%)			
No	89.7	88.4	85.2
Yes	10.3	11.6	14.8
**BMI (%)**			
Normal (<25 kg/m^2^)	55.8	45.9	43.0
Overweight (25–30 kg/m^2^)	35.7	43.3	42.5
Obese (>30 kg/m^2^)	8.5	10.8	14.5
**Leisure-time physical activity** (%)			
At least twice a week	32.9	29.9	24.7
Once a week	45.7	47.2	48.0
Less frequently	21.4	23.0	27.4
**Sleep problems** (Mean, SD)	3.3 (2.1)	4.2 (2.4)	4.6 (2.4)
**Morbidity and injuries** (%)			
Musculoskeletal (yes vs. no)	31.1	57.7	64.8
Cardiovascular (yes vs. no)	14.1	22.5	32.2
Respiratory (yes vs. no)	8.9	17.5	18.9
Mental (yes vs. no)	3.4	5.7	10.1
Other diseases (yes vs. no)	19.0	32.7	38.3
Injuries (yes vs. no)	7.8	15.7	20.3
**Working conditions** (Mean, SD)			
Awkward work postures	5.0 (3.0)	6.3 (2.8)	6.6 (2.6)
Physical demands	4.7 (2.7)	5.8 (2.6)	5.8 (2.6)
Physical and chemical exposures	1.7 (1.6)	2.2 (1.8)	2.3 (1.8)
Satisfaction with management	3.2 (1.8)	3.4 (2.9)	3.4 (1.9)
Job control	5.5 (2.4)	4.7 (2.5)	4.7 (2.4)
Mental demands	5.3 (3.0)	4.4 (3.0)	4.3 (3.3)
**b. Men**			
**Characteristic**	**Sickness Absence Trajectory**		
	**Low** **(n = 2028)**	**Intermediate** **(n = 491)**	**Increasing** **(n = 278)**
**Age** (Mean, SD)	50.2 (3.6)	50.8 (3.7)	51.7 (3.5)
**Basic education** (%)			
Elementary school not completed	26.3	28.5	38.0
Elementary school completed	56.5	63.8	54.5
Complementary school completed	17.3	7.6	7.7
**Family income** (%)			
Good	30.7	20.6	20.5
Satisfactory	64.2	72.2	73.3
Poor	5.2	7.2	6.2
**Lifestyle factors**			
**Smoking** (%)			
No	74.1	60.5	60.5
Yes	25.9	39.5	39.5
**BMI** (%)			
Normal (<25 kg/m^2^)	38.2	35.4	35.6
Overweight (25–30 kg/m^2^)	52.2	53.1	50.9
Obese (>30 kg/m^2^)	9.6	11.5	13.5
**Leisure-time physical activity** (%)			
At least twice a week	30.3	21.1	20.0
Once a week	48.0	52.9	47.4
Less frequently	21.7	25.9	32.6
**Sleep problems** (Mean, SD)	3.1 (2.1)	4.0 (2.4)	4.4 (2.3)
**Morbidity and injuries** (%)			
Musculoskeletal (yes vs. no)	28.4	50.1	54.7
Cardiovascular (yes vs. no)	18.7	24.2	32,0
Respiratory (yes vs. no)	7.8	14.7	14.0
Mental (yes vs. no)	3.7	8.2	6.1
Other disease (yes vs. no)	24.9	38.7	42.5
Injuries (yes vs. no)	13.4	20.8	20.5
**Working conditions**			
Awkward work postures	4.4 (2.9)	5.7 (2.6)	5.6 (2.9)
Physical demands	3.6 (2.5)	4.1 (2.5)	4.6 (2.6)
Physical and chemical exposures	3.1 (2.4)	4.0 (2.3)	3.9 (2.2)
Satisfaction with management	3.4 (1.9)	3.6 (2.0)	3.6 (1.9)
Job control	4.8 (2.6)	4.2 (2.4)	4.2 (2.4)
Mental demands	5.6 (2.9)	5.4 (2.9)	5.2 (2.8)

**Table 2 ijerph-18-02614-t002:** Associations of lifestyle factors, morbidity, injuries, and working conditions at baseline with sickness absence trajectories over a 5-year period (1981–1985). Multinomial logistic regression analysis. Model 1: adjusted for age. Model 2: mutually adjusted for all variables in the table, age, and socioeconomic status ^1^. Women.

	Model 1				Model 2			
	Intermediate vs. Low		Increasing vs. Low		Intermediate vs. Low		Increasing vs. Low	
	OR	95% CI	OR	95% CI	OR	95% CI	OR	95% CI
*Lifestyle factors*								
**Smoking (yes vs. no)**	1.15	0.84–1.59	0.71	0.28–1.80	1.16	0.82–1.64	0.55	0.21–1.46
**BMI, kg/m^2^ (ref: < 25)**								
25-30	1.55	1.24–1.93	1.47	0.84–2.59	1.31	1.03–1.65	1.13	0.63–2.04
≥30	1.71	1.21–2.43	3.79	1.97–7.31	1.26	0.86–1.85	2.13	1.03–4.40
**LTPA ^2^ (ref: At least twice a week)**								
Once a week	1.18	0.93–1.50	1.84	0.93–3.61	1.04	0.81–1.34	1.62	0.81–3.27
Less frequently	1.13	0.85–1.50	2.84	1.40–5.76	0.97	0.71–1.32	2.40	1.12–5.11
**Sleep problems (ref: no)**								
Moderate	1.35	1.03–1.76	1.17	0.56–2.45	1.11	0.84–1.47	0.92	0.43–1.98
Frequent	2.58	2.02–3.28	3.84	2.14–6.89	1.74	1.33–2.27	2.08	1.10–3.96
*Morbidity and injuries*								
**Musculoskeletal (yes vs. no)**	3.15	2.55–3.89	5.98	3.44–10.4	2.17	1.72–2.73	3.58	1.96–6.54
**Cardiovascular (yes vs. no)**	1.86	1.44–2.41	2.17	1.23–3.84	1.33	1.00–1.76	1.26	0.68–2.33
**Respiratory (yes vs. no)**	2.52	1.89–3.36	3.96	2.23–7.03	1.73	1.27–2.36	2.81	1.50–5.27
**Mental (yes vs. no)**	1.49	0.89–2.51	5.16	2.45–10.9	0.91	0.53–1.59	2.52	1.07–5.91
**Other diseases (yes vs. no)**	2.26	1.80–2.84	2.27	1.34–3.84	1.52	1.18–1.95	1.17	0.66–2.09
**Injuries (yes vs. no)**	2.03	1.48–2.79	3.44	1.86–6.36	1.28	0.90–1.81	2.02	1.03–3.95
*Working conditions*								
**Awkward work postures (ref: low)**								
Moderate	1.97	1.49–2.59	1.79	0.90–3.55	1.35	0.98–1.86	1.28	0.57–2.87
High	2.95	2.27–3.84	3.05	1.63–5.69	1.55	1.10–2.19	1.45	0.63–3.35
**Physical demands (ref: low)**								
Moderate	1.87	1.44–2.44	1.26	0.66–2.41	1.29	0.94–1.77	0.80	0.37–1.73
High	2.73	2.09–3.57	2.51	1.38–4.57	1.41	0.98–2.02	1.22	0.53–2.80
**Physical and chemical exposures (ref: low)**								
Moderate	1.43	1.11–1.85	1.56	0.85–2.87	1.09	0.83–1.44	1.03	0.53–1.97
High	2.00	1.55–2.57	1.83	0.99–3.38	1.04	0.77–1.39	0.78	0.39–1.58
**Satisfaction with management (ref: low)**								
Moderate	0.75	0.57–0.99	1.25	0.66–2.34	0.81	0.60–1.09	0.86	0.43–1.70
High	0.81	0.64–1.03	0.57	0.32–1.02	0.96	0.73–1.26	0.77	0.40–1.49
**Job control (ref: low)**								
Moderate	0.76	0.59–0.98	0.71	0.38–1.34	0.83	0.63–1.10	0.53	0.26–1.11
High	0.57	0.45–0.73	0.54	0.31–0.96	0.87	0.65–1.17	1.02	0.52–2.04
**Mental demands (ref: low)**								
Moderate	0.56	0.44–0.71	0.63	0.36–1.10	0.81	0.61–1.07	0.86	0.45–1.65
High	0.49	0.38–0.64	0.49	0.26–0.94	0.63	0.46–0.88	0.62	0.28–1.40

^1^ basic education and family income; ^2^ leisure-time physical activity.

**Table 3 ijerph-18-02614-t003:** Associations of lifestyle factors, morbidity, injuries, and working conditions at baseline with sickness absence trajectories over a 5-year period (1981–1985). Multinomial logistic regression analysis. Model 1: adjusted for age. Model 2: mutually adjusted for all variables in the table, age, and socioeconomic status ^1^. Men.

	Model 1				Model 2			
	Intermediate vs. Low		Increasing vs. Low		Intermediate vs. Low		Increasing vs. Low	
	OR	95% CI	OR	95% CI	OR	95% CI	OR	95% CI
*Lifestyle factors*								
**Smoking (yes vs. no)**	1.82	1.44–2.30	1.93	1.29–2.92	1.64	1.27–2.11	1.61	1.03–2.50
**BMI, kg/m^2^ (ref: < 25)**								
25-30	1.16	0.91–1.47	1.13	0.73–1.73	1.09	0.84–1.41	1.05	0.66–1.65
≥30	1.41	0.96–2.07	1.50	0.76–2.95	1.33	0.88–2.01	1.38	0.66–2.86
**LTPA ^2^ (ref: at least twice a week)**								
Once a week	1.46	1.10–1.93	2.01	1.12–3.63	1.19	0.88–1.61	1.58	0.85–2.97
Less frequently	1.55	1.12–2.14	3.42	1.86–6.30	1.11	0.78–1.58	2.35	1.20–4.60
**Sleep problems (ref: no)**								
Moderate	1.40	1.06–1.86	2.05	1.20–3.52	1.17	0.87–1.58	1.53	0.87–2.70
Frequent	2.36	1.81–3.08	4.32	2.66–7.00	1.51	1.11–2.03	2.30	1.35–3.93
*Morbidity and injuries*								
**Musculoskeletal (yes vs. no)**	2.49	1.97–3.13	3.53	2.36–5.28	1.70	1.31–2.20	2.28	1.45–3.57
**Cardiovascular (yes vs. no)**	1.32	1.00–1.73	2.06	1.34–3.17	0.94	0.70–1.26	1.33	0.84–2.11
**Respiratory (yes vs. no)**	1.97	1.40–2.78	2.64	1.53–4.55	1.31	0.89–1.91	1.42	0.78–2.57
**Mental (yes vs. no)**	2.52	1.59–4.00	3.03	1.44–6.35	1.80	1.09–2.95	1.68	0.76–3.71
**Other diseases (yes vs. no)**	2.11	1.67–2.67	2.18	1.45–3.27	1.53	1.18–1.98	1.29	0.83–2.02
**Injuries (yes vs. no)**	1.72	1.29–2.32	1.39	0.82–2.37	1.11	0.80–1.53	0.80	0.45–1.41
*Working conditions*								
**Awkward work postures (ref: low)**								
Moderate	2.50	1.85–3.37	1.49	0.88–2.53	1.80	1.26–2.57	0.81	0.44–1.52
High	3.42	2.51–4.67	3.25	1.97–5.36	1.92	1.29–2.86	1.25	0.65–2.41
**Physical demands (ref: low)**								
Moderate	1.19	0.90–1.59	1.61	0.94–2.77	0.97	0.71–1.32	1.37	0.77–2.42
High	1.83	1.39–2.39	2.66	1.60–4.41	1.07	0.78–1.47	1.62	0.90–2.91
**Physical and chemical exposures (ref: low)**								
Moderate	1.85	1.38–2.47	2.58	1.54–4.33	1.20	0.86–1.66	1.70	0.96–3.04
High	2.75	2.06–3.67	2.94	1.71–5.04	1.32	0.92–1.88	1.22	0.63–2.36
**Satisfaction with management (ref: low)**								
Moderate	1.09	0.82–1.46	0.87	0.54–1.39	1.33	0.98–1.81	1.04	0.62–1.73
High	0.75	0.57–0.97	0.34	0.21–0.56	1.06	0.79–1.42	0.49	0.28–0.85
**Job control (ref: low)**								
Moderate	0.69	0.53–0.90	0.66	0.41–1.04	0.79	0.60–1.06	0.82	0.50–1.36
High	0.50	0.38–0.66	0.40	0.24–0.67	0.84	0.61–1.17	0.92	0.51–1.65
**Mental demands (ref: low)**								
Moderate	0.82	0.63–1.09	0.98	0.63–1.53	0.91	0.69–1.21	1.19	0.76–1.88
High	0.87	0.67–1.14	0.47	0.27–0.80	1.03	0.78–1.36	0.60	0.34–1.03

^1^ basic education and family income; ^2^ leisure-time physical activity.

**Table 4 ijerph-18-02614-t004:** Membership in the 5-year sickness absence trajectories (1981–1985) as a predictor of disability pension retirement over a 10-year (1986–1996) follow-up. Cox regression analysis. Hazard ratios (HZ) and their 95% confidence intervals (CI) adjusted for age (model 1) and age and socioeconomic status ^1^ (model 2).

Sickness Absence Trajectories 1981–1985	Model 1		Model 2	
	HR	95% CI	HR	95% CI
**Women**				
Low	1	-	1	-
Intermediate	3.25	2.79–3.77	3.00	2.58–3.50
Increasing	10.19	6.56-15.81	9.84	6.33-15.28
**Men**				
Low	1	-	1	-
Intermediate	2.32	1.97–2.73	2.19	1.85–2.58
Increasing	24.07	16.19–35.79	21.81	14.58–32.63

^1^ basic education and family income.

## Data Availability

The data presented in this study are available on request from the corresponding author.

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
