# Peer review of "Process of Work Disability: From Determinants of Sickness Absence Trajectories to Disability Retirement in a Long-Term Follow-Up of Municipal Employees"

_ijerph, 2021, doi:10.3390/ijerph18052614_

Round 1

Reviewer 1 Report

The authors present data about determinants of, and trajectories of, sickness absence lasting > 10 days amongst municipal workers over five years of follow-up and subsequent risk of disability pension over >15 years of follow-up. These analyses, although complex, are presented clearly. It is excellent to see health data alongside work characteristics.

  1. Table 1 shows very clearly the socio-economic gradient of sickness absence trajectories. Those with the increasing trajectory have the lowest educational attainment, poorer household income, are more likely obese, smokers, take less LTPA, and have more MSDs at baseline (and to a lesser extent cardio-respiratory diseases) and amongst, women, more mental health conditions at baseline. They also have the worst working conditions (at least by their own self-report) including low autonomy and higher physical loads. It was pleasing to see Model 1 in Tables 2 was only adjusted for age (so that the s/e status was not adjusted out initially). Although this socio-economic gradient is by no means new, it would be nice to see this point brought out a little more in the Discussion. We of course have known this but the sadness is that it does not seem to be changing with more contemporary data. Moreover, if Finland, like other European countries wants to retain older workers in the workplace to older ages, then this socio-economic gradient will mean that those with poorest physical health, starting from poorest backgrounds are going to be least likely to work to older ages. We need the public health agenda around smoking, obesity and LTPA to include messages about work ability.
  2. The Kaplan Meier plots for DP are really interesting and useful. They starkly show the relevance of increasing sickness absence to DP in Finland. It is of course not unexpected that more days sick leave lead to DP especially when the criterion for DP is >300 days of sick leave and it is worth bringing out more clearly in the Discussion that the relationship might be less clear cut in other countries where such a  system does not apply. In the UK, for example, most people are not eligible for a DP unless they or their employer voluntarily pays an insurance to cover that necessity (therefore this is more likely amongst more valued, better paid workers). Therefore, if an individual becomes unfit for work, they are either reliant upon personal wealth or state welfare disability benefits until they are eligible for state pension.
  3. A small pint but the alignment in table 1 had gone wrong in my version, making it rather difficult to read across.

Author Response

Thank you very much for the positive feedback.

Reviewer #1

The authors present data about determinants of, and trajectories of, sickness absence lasting > 10 days amongst municipal workers over five years of follow-up and subsequent risk of disability pension over >15 years of follow-up. These analyses, although complex, are presented clearly. It is excellent to see health data alongside work characteristics.

Response: Thank you very much for the positive feedback.

  1. Table 1 shows very clearly the socio-economic gradient of sickness absence trajectories. Those with the increasing trajectory have the lowest educational attainment, poorer household income, are more likely obese, smokers, take less LTPA, and have more MSDs at baseline (and to a lesser extent cardio-respiratory diseases) and amongst, women, more mental health conditions at baseline. They also have the worst working conditions (at least by their own self-report) including low autonomy and higher physical loads. It was pleasing to see Model 1 in Tables 2 was only adjusted for age (so that the s/e status was not adjusted out initially). Although this socio-economic gradient is by no means new, it would be nice to see this point brought out a little more in the Discussion. We of course have known this but the sadness is that it does not seem to be changing with more contemporary data. Moreover, if Finland, like other European countries wants to retain older workers in the workplace to older ages, then this socio-economic gradient will mean that those with poorest physical health, starting from poorest backgrounds are going to be least likely to work to older ages. We need the public health agenda around smoking, obesity and LTPA to include messages about work ability.

Response: Thank you very much for this positive feedback and the suggestion to underline the socioeconomic gradient. We have now added some comments on it in the discussion and emphasized it in the conclusion at the end of it.

  1. The Kaplan Meier plots for DP are really interesting and useful. They starkly show the relevance of increasing sickness absence to DP in Finland. It is of course not unexpected that more days sick leave lead to DP especially when the criterion for DP is >300 days of sick leave and it is worth bringing out more clearly in the Discussion that the relationship might be less clear cut in other countries where such a  system does not apply. In the UK, for example, most people are not eligible for a DP unless they or their employer voluntarily pays an insurance to cover that necessity (therefore this is more likely amongst more valued, better paid workers). Therefore, if an individual becomes unfit for work, they are either reliant upon personal wealth or state welfare disability benefits until they are eligible for state pension.

Response: We first apologies for an error in the text of Figure 2: it plots Nelson-Aalen cumulative hazards instead of Kaplan-Meier estimates (we changed the figure but forgot to change the text under it). Thank you very much for the example from the UK. We now discuss shortly the matter of varying policies by country.

  1. A small pint but the alignment in table 1 had gone wrong in my version, making it rather difficult to read across.

Response: Thank you for pointing this out. This was a technical error when converting the table into the journal’s template. This has been corrected now.

Reviewer 2 Report

Dear authors,

Very intersting presentation of never ending problem.

There are some questions to be clarified -

The observed period goes back to 1981-1985 and 1986-1995. Is this period realy relevant to make conclusions to the current times? namely the proceedures of diagnostic, therapies and also working conditions has been changed very much since that times.

The statistical approach is relevant. But after all - is it possible to generalise the results of the study to European population? Sickness absence duration and verifying disability are very much determined by the legislation of the states - could you make any general conclusion for any European country? 

The prosposals are very shortly mentioned (line 167-168), it would be a better solution to make them more practical.

With best wishes i am looking forward to your answers.

Author Response

Thank you very much for the comments which we have considered carefully and revise the manuscript accordingly.

Reviewer #2

Very intersting presentation of neverending problem.

Response: Thank you very much for the positive feedback.

There are some questions to be clarified -

The observed period goes back to 1981-1985 and 1986-1995. Is this period realy relevant to make conclusions to the current times? namely the proceedures of diagnostic, therapies and also working conditions has been changed very much since that times.

Response: This is in fact one of the limitations of our study which we have discussed as limitation in the discussion. However, we also argue that the jobs in the municipal sector such as nursing, transport, administrative, kitchen, physician, teaching, etc. probably have not changed drastically since the start of this study.

The statistical approach is relevant. But after all - is it possible to generalise the results of the study to European population? Sickness absence duration and verifying disability are very much determined by the legislation of the states - could you make any general conclusion for any European country? 

Response: Our findings may not in detail be generalizable to all European countries due to differences in sickness absence legislation and other social benefits, as the reviewer points out. However, we suggest that analyzing how disability develops in one cohort from one country may give some useful insight on the long-drawn process, that could even be generalizable to a wider population.

The prosposals are very shortly mentioned (line 167-168), it would be a better solution to make them more practical.

Response: We have now provided few sentences in the discussion to discuss the policy implications of our study.

Reviewer 3 Report

The topic of the manuscript is interesting, but the paper needs several clarifications before it is suitable for publication. Below are some comments to help further improve and clarify the manuscript.  

  1. The abstract should give more information on the data: at least the country setting, register sources, study years (baseline year and follow-up periods), the age of the participants etc.
  2. There are a lot of previous studies addressing the determinants of sickness absence, as well as studies on the disability process from sickness absence to disability retirement. Thus, the authors should state clearly what is the rationale for their study in comparison to previous studies. What was the explicit lack of knowledge that this study addressed? This should be stated in the introduction and revisited in the discussion.
  3. More information on the study sample should be given in Chapter 2.1. For example: How exactly was the sample constructed? Was the sample collected from some municipalities only? Was it a random sample of some pre-defined group?
  4. The data are really old – the baseline questionnaire data were collected in 1980–1981. Sickness absence follow-up data was collected from years 1981–1985 and disability retirement data from 1986 to 1995. The authors should add some reflections and explanation for using a baseline data set collected as long as 40 years ago, with follow-up that ended 26 years ago. How can very old data help gain understanding that helps tackle today’s work disability problems? Working life and general morbidity issues, as well as the basic characteristics of municipality workers - such as their educational level - have greatly changed in 40 years.
  5. A descriptive table giving the distributions of all covariates for the total sample (at the minimum, columns for all men and all women) should be added before going to the results.
  6. To increase readability, the covariates should be presented in Chapter 2 in the same order in the text as they are shown later in the tables.
  7. Some covariates are described differently in Chapter 2 compared to how they are presented in, for example, Table 1. For example, on sleep problems, the text states that a sleep problem score was formed and then categorized into tertiles, however the table seems to show a score. There are also other inconsistencies. Please crosscheck all variable descriptions, category names etc. in the text and in the tables.
  8. Table 1 is difficult to read and understand as some rows or columns seem to be misplaced. Layout of the table should be improved and streamlined.
  9. After the baseline, sickness absence was assessed from 1981 to 1985 and then disability retirement from 1986 to 1995. The authors should add some description in chapter 2.8 (Statistical Methods) on how they treated persons who retired or died after the baseline but before the latter follow-up period, i.e. during the period of sickness absence assessment.
  10. In the figures, women are presented first, whereas in the tables, men are presented first. Please be consistent.
  11. In Figure 1, what do the “expected” lines mean? The analyses leading to the “expected” numbers should be explained. Also, the tick mark labels of the X axis should be years rather than days if the sickness absence measurements were yearly as stated in the text.
  12. Results in Table 1 are not commented in the text at all. Is the table redundant? At least some sentences should be added in the text if the table is included.
  13. Tables 2A and 2B are very difficult to read because of layout problems concerning the first column that includes variable names and categories. Please correct this. Also, as there are separate tables for men (Table 2A) and women (2B), the text should clearly refer to these when quoting the results.
  14. Also concerning reporting of results in Tables 2A and 2B, it would be better to organize the text and the Tables in the same order, as concerns presentation of results of different variables.
  15. It is unnecessary to repeat almost all statistically significant results with their confidence intervals in the text, since they are shown in the tables. Reporting of these results could be somewhat streamlined.
  16. There should be more description of the disability retirement variable before going to more detailed analyses. On page 12, it is stated that there were 1782 disability retirement cases. Does that mean that as many as 28% (1728/6257) of the study cohort experienced disability retirement during the 10-year follow-up? This knowledge could be given in the text and/or included in a table, and also the proportions of retirees in the trajectory subgroups. When comparing the given numbers of disability retirees in each of the sickness absence trajectory groups (page 12, lines 53-54) to the group sizes given in Table 1, it seems that in relative terms, among those with an increasing sickness absence trajectory, the proportion of disability retirees during the follow-up was clearly lowest. How is this possible given the rather opposite results shown later on page 12, lines 56-62 and in the Kaplan-Meier figures?
  17. The Kaplan-Meier estimates shown in Figure 3 should be interpreted more clearly. According to estimates shown in the Figures, the probabilities of disability retirement during the follow-up seem to be very low (less than 2.5% in any of the groups); is that a correct interpretation? Here, the authors could explain their results more.
  18. The labels of the x axis in Figure 3 should be corrected. The measurement of disability retirement seems to be yearly so the labels should not show half-years. Why do the last persons in Figure 3 enter retirement at around 14 years of follow up - no-one at 15 years? Why are the Kaplan-Meier curves so short for the increasing SA trajectory?
  19. A Table 3 appears at the end of the Discussion, with no previous reference to this table in the text. This table should be moved to the correct place in the Results chapter, and the reporting of the results should refer to this table on page 12, lines 59-61.
  20. In the discussion, some elaboration is needed on whether these results, largely derived from a data set collected around 40 years ago, are relevant today and what may have changed in the system and associations. What results are so novel and pertinent that they need to be published even though the data are so old? What are the policy implications of the study?
  21. The formatting of the manuscript should be corrected to follow the guidelines of the Journal, for example concerning formatting of tables, references and the reference list.

Author Response

Thank you very much for the comments which we have considered carefully and revise the manuscript accordingly. Please find attached the point by point response to the comments in a separate word document.

Reviewer 4 Report

Dear Authors,

I found your study interesting and well- written. Crossing perception and observational data in a longitudinal perspective is a great value added, particularly in the view of studying the work disability process. Thi requires to follow workers over a great number of years and this is not easy.

Nevertheless, I think your study suffers of 2 main weakness.

First of all there is not any questionnaire follow up, thus we cannot appreciate eventual changes in working conditions perceptions and lifestyles of the participants. We cannot know if they changed jobs in the next 15 years (and in case working conditions) or if they changed their lifestyles (in some cases this could happen also considerably as quitting smoking, starting exercising, doing a healthy diet).

The second concern is that we must recognized that data used is old. I completely understand the difficulties in collecting such kind of data, but we are referring completely to another world of work. Working conditions has now changed. We are into the fourth industrial revolution and workforce is characterized strongly for ageing.

Since I recognized the difficulties in having access to prospective data, I feel the article could give a contribution to the scientific community on the topic of the determinants of disability.

Nevertheless, I think authors must do an effort in contextualizing the study in the current period (also updating more the references). This can be done in the introduction and discussion, by opening a reflection on relevant nowadays aspects that cannot be analysed in this study or how your data and results can offer a contribution today. Moreover, weaknesses must be clearly highlighted offering reflections for future perspectives.

Finally, implications for practices could be reinforced a bit. Only a sentence refers to possible interventions.

Author Response

Thank you very much for the comments which we have considered carefully and revise the manuscript accordingly.

Reviewer #4

I found your study interesting and well- written. Crossing perception and observational data in a longitudinal perspective is a great value added, particularly in the view of studying the work disability process. Thi requires to follow workers over a great number of years and this is not easy.

Response: Thank you very much for the positive feedback.

Nevertheless, I think your study suffers of 2 main weakness.

First of all there is not any questionnaire follow up, thus we cannot appreciate eventual changes in working conditions perceptions and lifestyles of the participants. We cannot know if they changed jobs in the next 15 years (and in case working conditions) or if they changed their lifestyles (in some cases this could happen also considerably as quitting smoking, starting exercising, doing a healthy diet).

Response: Yes, we have not provided data from questionnaire follow-up for the changes in the working conditions. However, as there indeed were several follow-ups of the survey, we have added information based on our earlier study using the same cohort. It showed a relative stability of working conditions and lifestyle during 11 years of follow-up from the baseline survey (Neupane et al, 2018).

The second concern is that we must recognized that data used is old. I completely understand the difficulties in collecting such kind of data, but we are referring completely to another world of work. Working conditions has now changed. We are into the fourth industrial revolution and workforce is characterized strongly for ageing.

Response: This is in fact one of the limitations of our study. This has been discussed in the limitations section where we reflect the usefulness of old data in the present context. However, we argue that many jobs in the municipal sectors such as nursing, transport, administrative, kitchen, teaching, etc. probably have not changed drastically since the start of this study. Moreover, studying the process of disability from the development of sickness absence till retirement requires a stretch of time, and we feel that there may be elements that can be considered of value even today.

Since I recognized the difficulties in having access to prospective data, I feel the article could give a contribution to the scientific community on the topic of the determinants of disability.

Response: Thank you very much for your compliment.

Nevertheless, I think authors must do an effort in contextualizing the study in the current period (also updating more the references). This can be done in the introduction and discussion, by opening a reflection on relevant nowadays aspects that cannot be analysed in this study or how your data and results can offer a contribution today. Moreover, weaknesses must be clearly highlighted offering reflections for future perspectives.

Response: We thank the reviewer for this suggestion. We have now provided more references in the introduction and in discussion to reflect the contribution of our data in today’s context.

Finally, implications for practices could be reinforced a bit. Only a sentence refers to possible interventions.

Response: We have now provided few sentences in the discussion to discuss the policy implications of our study.

Reference

Neupane, S., Nygård, C. H., Prakash, K. C., von Bonsdorff, M. B., von Bonsdorff, M. E., Seitsamo, J., ... & Leino-Arjas, P. (2018). Multisite musculoskeletal pain trajectories from midlife to old age: a 28-year follow-up of municipal employees. Occupational and Environmental Medicine75(12), 863-870.

Round 2

Reviewer 3 Report

The authors have replied to and corrected some of the points raised and the manuscript has improved from the previous version. Unfortunately, not all points in my previous review have been replied to. Thus, below I repeat some points from the previous review that need to be revisited (the numbering refers to the points in the first review). Please, when replying, give the specific page and line numbers on where the changes can be found in the manuscript.

Point #1: Abstract

  • Relevant information on the data has been added; however, the added info that retirement and mortality were followed until 2009 is misleading, as the analyses extended only until year 1995. The year should be corrected in the abstract.

Point #6: The presentation order of covariates should be streamlined to increase readability:

  • In the methods chapter, the order of the variable blocks is: 1) morbidity, 2) age & socio-demographics, 3) working conditions, 4) lifestyle factors. Why not present the blocks in the same order as they appear in the Tables: (1) age & socio-demographics, 2) lifestyle factors, 3) morbidity, 4) working conditions)

Point #7: Please crosscheck all variable descriptions, category names etc. in the text and in the tables.

  • This has not been corrected. For example, compare Tables 1 and 2 and ensure that variable names and categories are consistent across the total manuscript (including the variable descriptions in the methods section). Some examples: Naming of BMI and LTPA categories are different in Tables 1 and 2. Smoking is missing from Table 1 but included in Table 2; thus it must be added also in Table 1 with the other variables. Crosscheck variables in all instances.

Point #11: The “expected” lines in Figure 1.

  • The authors should explain in the text what the expected lines mean in Figure 1 (the label for the line was already there in the Figure in the first version, but still no explanation has been added in the text).

Point # 11: Tick mark labels.

  • This has not been corrected nor answered to. The tick mark labels of the X axis in Figure 1 should be years rather than days, since sickness absence was measured at a yearly basis (not daily) as stated in the text.

Point # 16: Disability retirement in sickness absence subgroups.

  • The authors did not explain why disability retirement proportions according to SA trajectories are opposite when calculated from crude numbers than in the modeling results. My original question was: “When comparing the given numbers of disability retirees in each of the sickness absence trajectory groups (...) to the group sizes given in Table 1, it seems that in relative terms, among those with an increasing sickness absence trajectory, the proportion of disability retirees during the follow-up was clearly lowest. How is this possible?“ Please provide a clear explanation.
  • Note that the text still mentions “Kaplan-Meier curves” in chapter 3.3 even though this should have been changed.

Point # 18: Tick marks in Figure 2 (sorry, I had mistakenly previously called this Figure 3) should be corrected.

  • This has not been done nor commented on. Please correct the tick marks to full years (5, 6, 7,…15; not intervals of 2.5 years) to increase clarity, as your measurement was yearly. The tick marks should be very easy to correct in both Figures.

Author Response

Reviewer #3

The authors have replied to and corrected some of the points raised and the manuscript has improved from the previous version. Unfortunately, not all points in my previous review have been replied to. Thus, below I repeat some points from the previous review that need to be revisited (the numbering refers to the points in the first review). Please, when replying, give the specific page and line numbers on where the changes can be found in the manuscript.

Response: Thank you very much for the careful reading and the comments and feedback on the revised version of the manuscript. We have now carefully looked at all the comments and revised the manuscript accordingly. As suggested, the specific page and line numbers are provided in the response below.

Point #1: Abstract

  • Relevant information on the data has been added; however, the added info that retirement and mortality were followed until 2009 is misleading, as the analyses extended only until year 1995. The year should be corrected in the abstract.

Response: We have now revised the text as suggested. The text now reads “….. were linked with registers on SA (>10 workdays), and disability pension from the period 1986-1995”. Please see the corrected text in the abstract, page 1, line 17-18

Point #6: The presentation order of covariates should be streamlined to increase readability:

  • In the methods chapter, the order of the variable blocks is: 1) morbidity, 2) age & socio-demographics, 3) working conditions, 4) lifestyle factors. Why not present the blocks in the same order as they appear in the Tables: (1) age & socio-demographics, 2) lifestyle factors, 3) morbidity, 4) working conditions)

Response: Thank you for this suggestion. Now we have revised the presentation order of covariates in same order as of Table 1 for readability. Please see the changes made in the manuscript in page 3-4.

Point #7: Please crosscheck all variable descriptions, category names etc. in the text and in the tables.

  • This has not been corrected. For example, compare Tables 1 and 2 and ensure that variable names and categories are consistent across the total manuscript (including the variable descriptions in the methods section). Some examples: Naming of BMI and LTPA categories are different in Tables 1 and 2. Smoking is missing from Table 1 but included in Table 2; thus it must be added also in Table 1 with the other variables. Crosscheck variables in all instances.

Response: Thank you very much for careful reading. We have now crosschecked all the inconsistencies and revised them in the manuscript text and in table. Please see the revised text in Table 2a-b. We also did not notice that smoking variable was missing from Table 1, which we have now added.

Point #11: The “expected” lines in Figure 1.

  • The authors should explain in the text what the expected lines mean in Figure 1 (the label for the line was already there in the Figure in the first version, but still no explanation has been added in the text).

Response: We have now provided few texts describing what expected lines means in Figure 1. The added text reads “The expected lines are the weighted average of all possible lines of the SA, with the corresponding probabilities used as weights. The expected trajectories were similar to those of observed trajectories for both women and men.”, in page 5, line 231-234.

Point # 11: Tick mark labels.

  • This has not been corrected nor answered to. The tick mark labels of the X axis in Figure 1 should be years rather than days, since sickness absence was measured at a yearly basis (not daily) as stated in the text.

Response: We have now changed the days into the years in the x-axis of Figure 1 as per suggestion.

Point # 16: Disability retirement in sickness absence subgroups.

  • The authors did not explain why disability retirement proportions according to SA trajectories are opposite when calculated from crude numbers than in the modeling results. My original question was: “When comparing the given numbers of disability retirees in each of the sickness absence trajectory groups (...) to the group sizes given in Table 1, it seems that in relative terms, among those with an increasing sickness absence trajectory, the proportion of disability retirees during the follow-up was clearly lowest. How is this possible?“ Please provide a clear explanation.
  • Note that the text still mentions “Kaplan-Meier curves” in chapter 3.3 even though this should have been changed.

Response: Thank you very much for careful reading. We actually forgot to respond this particular question earlier. Now, we have carefully checked our data and found that there was some mismatch in the number of disability pension by trajectory group which we have corrected now in the manuscript. The number of disability pension in increasing trajectory is still the same, however the percentage of those people who were followed after 1985 until 1995 were the highest in increasing trajectory group. This is because many people in the increasing trajectory group retired due to disability or other reason or died already before the follow-up starts. Therefore, the percentage calculated based on the number of people who were followed was highest in increasing trajectory although the number seems to be small compared to other trajectories.

This was indeed a mistake, now we have corrected the Kaplan-Meier curves to Nelson-Aalen curve. Please see the changes made in text in section 3.3.

Point # 18: Tick marks in Figure 2 (sorry, I had mistakenly previously called this Figure 3) should be corrected.

  • This has not been done nor commented on. Please correct the tick marks to full years (5, 6, 7,…15; not intervals of 2.5 years) to increase clarity, as your measurement was yearly. The tick marks should be very easy to correct in both Figures.

Response: We thanks the reviewer for this suggestion. Now we have replaced the old graphs by new one new distribution of disability pension by trajectory. This has also changed the estimates in the Table 3.